# TOWARDS FEW-SHOT ADAPTATION FOR DENSE CROSS-MODALITY IMAGE MATCHING

## ABSTRACT

Cross-modality image matching aims to establish correspondences between images captured under different sensing modalities. Recent advances in transformer-based dense matchers and large-scale synthetic training data have led to foundation models with strong generalization to unseen modalities. However, their performance degrades when the target modality diverges substantially from the pretraining distribution, making domain-specific adaptation essential. Since annotated data is often costly and limited, while unlabelled data is plentiful, we address this challenge by adapting pretrained dense matchers with a combination of few-shot labelled and abundant unlabelled samples. Specifically, we exploit the multi-scale architecture of dense matchers by using the finest-scale predictions to guide learning at coarser scales on unlabelled data. Extensive experiments across diverse modalities demonstrate that our approach consistently outperforms both foundation models and purely supervised adaptation, achieving up to 40% improvement in matching accuracy.

## 1 INTRODUCTION

Cross-modality image matching, which aims to identify corresponding point pairs between images captured under different modalities, is a fundamental task in computer vision. It underpins applications such as image alignment, information fusion, and change detection. The task is challenging because image pairs can exhibit not only geometric variations but also drastic appearance changes due to different imaging principles.

Traditional methods (Li et al., 2023) typically follow the "keypoint detection–descriptor extraction–matching" pipeline using SIFT-like features. In contrast, modern approaches (Tuzcuoğlu et al., 2024) adopt a learning-based paradigm, leveraging deep neural networks to directly learn optimal matches. Recent progress has been driven by transformer-based dense matchers (Edstedt et al., 2024) and large-scale synthetic cross-modality datasets (He et al., 2025; Ren et al., 2025), leading to the development of foundation models for cross-modality image matching. Foundation models such as MatchAnything (He et al., 2025) and MINIMA (Ren et al., 2025) demonstrate impressive generalization across diverse domains. However, as shown in Fig. 1, their performance degrades substantially on modalities that deviate significantly from their pretraining data, highlighting the need for domain-specific adaptation.

While labelled data is indispensable for training high-quality deep models, it is often expensive and time-consuming to obtain, especially when expert annotation is required. By contrast, unlabelled data is typically abundant and easily accessible. Semi-supervised learning (SSL) offers a promising solution by leveraging unlabelled data to learn generalizable representations, while using a small set of labelled samples to guide task-specific adaptation.

Although SSL has been extensively studied in classification (Yang et al., 2022), its application to cross-modality image matching remains largely unexplored. Existing efforts are limited to specific domains and mostly follow detector-based matching pipelines (Hughes & Schmitt, 2019; Liu et al., 2022), with little success in the extremely low-label regime. Our work addresses this gap by adopting the detector-free dense matching paradigm, which has demonstrated superior accuracy, and by introducing an SSL framework that works effectively with few-shot labelled samples across diverse modalities.

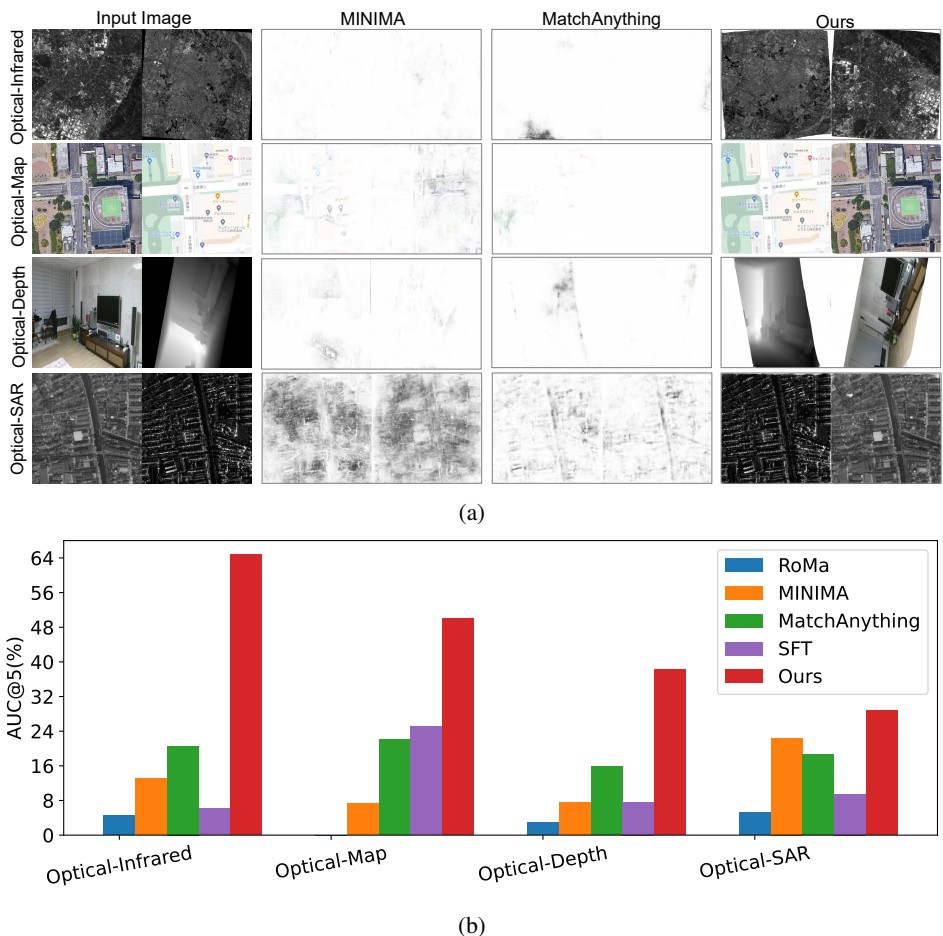

Figure 1: State-of-the-art foundation models perform poorly on modalities that differ significantly from their pretraining data. (a) Visualization of matching results. The results are visualized by transferring pixel values from the paired image using the estimated correspondences, weighted by the estimated certainty (white color indicates low certainty). (b) Quantitative comparison across different cross-modality matching methods. With only 2-shot labelled samples, our method improves $AUC$@5px by up to 40% relative to foundation models (see Section 4.2 for more details).

The central challenge of SSL is how to compute losses on unlabelled data. A common strategy is pseudo-labelling, which uses high-confidence predictions of the model as pseudo ground truth for training. In classification, pseudo-labels can be obtained by discretizing the predicted probability distribution into one-hot labels. However, dense matching involves not only classification but also regression tasks, which are typically optimized via an $\ell_p$-norm loss. Directly using model outputs as pseudo-labels for $\ell_p$-norm loss results in zero gradient during backpropagation, due to the lack of discretization step in regression tasks. One solution is to employ a separate teacher model, often an EMA (exponential moving averaging) of the student, to provide pseudo-labels. This approach will however incur additional memory and computation cost, and it is not guaranteed that the teacher model will always generate better pseudo labels than the student model. In this work, we instead exploit the multi-scale architecture of dense matchers to generate pseudo-labels.

Our main contributions are as follows:

- We propose a novel SSL framework for few-shot adaptation of dense matchers to downstream cross-modality data. Our method leverages the multi-scale architecture of dense matchers by using the finest-scale predictions as pseudo-labels to supervise coarser-scale learning.

- We investigate key design choices of the proposed pseudo-labelling method and identify effective pseudo-label generation and thresholding schemes for cross-modality image matching.
- We extensively evaluate the proposed method across diverse modalities and dataset difficulties, and demonstrate that our approach consistently improves matching accuracy over both purely supervised learning and foundation models.

## 2 RELATED WORK

### 2.1 VISIBLE IMAGE MATCHING

Visible image matching aims to identify corresponding pixel locations between images captured by visible cameras. Existing methods can be broadly categorized into three paradigms: detector-based (a.k.a. sparse) (Lowe, 2004; DeTone et al., 2018; Sarlin et al., 2020), detector-free (a.k.a. semi-dense) (Sun et al., 2021), and dense matching (Edstedt et al., 2023; 2024). Detector-based methods first detect a sparse set of keypoints, then extract features and perform matching on these locations. Traditional approaches, exemplified by the classical SIFT algorithm (Lowe, 2004), rely on handcrafted features for keypoint detection and descriptor extraction. Modern approaches employ deep neural networks for joint keypoint detection and description (DeTone et al., 2018; Sarlin et al., 2020). Detector-free methods, represented by LoFTR (Sun et al., 2021), eliminate the explicit keypoint detection step and instead match features at a coarse resolution, followed by refinement of the high-confidence coarse matches using fine-scale features. Dense matching methods, such as DKM (Edstedt et al., 2023) and RoMa (Edstedt et al., 2024), have recently demonstrated strong robustness under challenging real-world conditions. These methods estimate a 2D flow field at coarse scales and refine it to original image resolution progressively at finer scales..

### 2.2 CROSS-MODALITY IMAGE MATCHING

Cross-modality image matching addresses the more challenging setting where two images are captured under different sensing modalities. Methods typically adapt visible image matching pipelines to handle the large appearance variations between modalities, and can also be grouped into detector-based (Li et al., 2023), detector-free (Tuzcuoğlu et al., 2024), and dense matching (Ren et al., 2025; He et al., 2025) paradigms. SRIF (Li et al., 2023) enhances structural information through a local intensity binary transform and computes SIFT-like descriptors on the transformed images. XoFTR (Tuzcuoğlu et al., 2024) extends LoFTR by introducing a fine-level matching module and a sub-pixel refinement stage. More recently, foundation models such as MINIMA (Ren et al., 2025) and MatchAnything (He et al., 2025) pretrain existing matching models (*e.g.*, LoFTR and RoMa) on large-scale synthetic cross-modality pairs, achieving strong generalization to unseen modalities.

### 2.3 SEMI-SUPERVISED LEARNING

Semi-supervised learning (SSL) improves model generalization by leveraging unlabelled data in addition to a limited set of labelled samples. Consistency regularization and pseudo-labeling are two popular paradigms for SSL. Consistency regularization is based on the smoothness assumption: model predictions should remain stable under realistic perturbations (Yang et al., 2022). This can be achieved through input perturbations or augmentations (Miyato et al., 2018; Xie et al., 2020), or by employing a teacher-student framework where a teacher model provides consistency targets (Tarvainen & Valpola, 2017). Pseudo-labelling (Lee et al., 2013) relies on the assumption that high-confidence predictions are likely correct and can serve as pseudo ground truth for unlabelled data. It can be combined with consistency regularization by enforcing consistent pseudo-labels across weakly and strongly augmented samples (Sohn et al., 2020). A fixed confidence threshold (*e.g.*, 0.95) is typically used to select reliable pseudo-labels, while recent methods (Zhang et al., 2021; Wang et al., 2022) adapt thresholds dynamically based on the model's learning status.

### 2.4 SEMI-SUPERVISED LEARNING FOR IMAGE MATCHING

Most SSL approaches for image matching have been developed within the detector-based paradigm. For example, Hughes & Schmitt (2019) proposed an SSL framework for Optical-SAR (Synthetic

Aperture Radar) matching by training an autoencoder with reconstruction loss on unlabelled data and matching loss on labelled data. SuperRetina (Liu et al., 2022) exploits unlabelled data for retinal image matching by enforcing consistent keypoint and descriptor predictions under geometric transformations. For dense visible image matching, Truong et al. (2021) proposed to warp one of the two images by a randomly sampled flow and derive an unsupervised loss by bipath consistency constraints. To the best of our knowledge, SSL has not yet been explored for dense cross-modality image matching. Our work addresses this gap by introducing the first SSL framework tailored for this setting.

## 3 METHOD

### 3.1 BACKGROUND: SUPERVISED LEARNING FOR DENSE MATCHERS

Given two images $I^{\mathcal{A}}$ and $I^{\mathcal{B}}$, dense matchers estimate a flow field by first performing global matching on coarse-scale features and then refining the warp on finer-scale features. Let $S$ denote the set of scales. During training, the loss is computed at each scale and summed to form the final objective: $\mathcal{L} = \sum_{i \in S} \mathcal{L}_i$. The loss is defined on normalized coordinates in $[-1, 1]$, where each scale $i$ predicts correspondences on a grid of size $h_i \times w_i$ (the height and width of the feature map). Let $X_i^{\mathcal{A}}$ denote the set of locations considered at scale $i$ in $I^{\mathcal{A}}$, and $X_i^{\mathcal{AP}} \subseteq X_i^{\mathcal{A}}$ the subset with reliable matches in $I^{\mathcal{B}}$. At each scale, the loss consists of a regression loss term (*e.g.*, $\ell_2$ loss in (Edstedt et al., 2023), generalized Charbonnier loss in (Edstedt et al., 2024)) on the warp and a binary cross-entropy (BCE) loss term on the certainty score:

$$\mathcal{L}_i^{\text{sup}} = \mathbb{E}_{(I^{\mathcal{A}}, I^{\mathcal{B}}) \in I_L} \left[ \frac{1}{|X_i^{\mathcal{AP}}|} \sum_{x \in X_i^{\mathcal{AP}}} \mathcal{L}_{reg}(\hat{w}(x), w(x)) + \lambda^{sup} \frac{1}{|X_i^{\mathcal{A}}|} \sum_{x \in X_i^{\mathcal{A}}} \mathcal{L}_{BCE}(\hat{p}(x), p(x)) \right], \tag{1}$$

where $I_L$ is the set of labelled image pairs, $\hat{w}(x)$ and $w(x)$ are the predicted and ground-truth matching coordinates in $I^{\mathcal{B}}$ for pixel $x$ in $I^{\mathcal{A}}$, and $\hat{p}(x)$ and $p(x)$ are the predicted and ground-truth certainty scores. The ground-truth warp $w(x)$ is obtained by projecting coordinates from $I^{\mathcal{A}}$ into $I^{\mathcal{B}}$ using the known transformation (*e.g.*, homography or camera pose). The certainty ground truth $p(x)$ is generated by applying visibility and depth consistency constraints on the projection (*i.e.*, pixels satisfying the constraints are classified as 1, otherwise 0).

### 3.2 SEMI-SUPERVISED LEARNING FOR DENSE MATCHERS

State-of-the-art dense matchers (Edstedt et al., 2023; 2024) refine predictions progressively across scales, with the finest-scale prediction providing the final output. We propose to use these high-confidence finest-scale predictions as pseudo-labels for supervising coarser scales. Specifically, the finest-scale warp and certainty maps, $\hat{w}^f(\cdot)$ and $\hat{p}^f(\cdot)$, are interpolated to the grid of scale $i$ via bilinear interpolation, yielding $\hat{w}_i^f(\cdot)$ and $\hat{p}_i^f(\cdot)$. This assumes local smoothness of the motion field, which generally holds since high-confidence predictions rarely occur at motion boundaries.

The SSL loss at scale $i$ is defined as:

$$\mathcal{L}_i^{\text{ssl}} = \mathbb{E}_{(I^{\mathcal{A}}, I^{\mathcal{B}}) \in I_U} \left[ \frac{1}{|\hat{X}_i^{\mathcal{AP}}|} \sum_{x \in \hat{X}_i^{\mathcal{AP}}} \mathcal{L}_{reg}(\hat{w}(x), \hat{w}_i^f(x)) + \lambda^{ssl} \frac{1}{|\hat{X}_i^{\mathcal{A}}|} \sum_{x \in \hat{X}_i^{\mathcal{A}}} \mathcal{L}_{BCE}(\hat{p}(x), \mathbb{I}(\hat{p}_i^f(x) > \tau_h)) \right], \tag{2}$$

where $I_U$ denote the set of image pairs without ground truth labels, $\mathbb{I}(\cdot)$ is the indicator function that takes 1 if the condition is met and 0 if otherwise, $\hat{X}_i^{\mathcal{AP}} = \{x | \hat{p}_i^f(x) > \tau_h\}$ and $\hat{X}_i^{\mathcal{A}} = \{x | \hat{p}_i^f(x) > \tau_h | \hat{p}_i^f(x) < \tau_l\}$.

Instead of using fixed thresholds for pseudo-labelling, we adopt a self-adaptive strategy inspired by FreeMatch (Wang et al., 2022). We maintain a global threshold $\tau_t$ reflecting the model's overall

confidence on unlabelled data:

$$\tau_t = \begin{cases} 0.5, & t = 0, \\ \gamma\tau_{t-1} + (1-\gamma)\frac{1}{|X_f^b|}\sum_{x \in X_f^b} \hat{p}^f(x), & \text{otherwise,} \end{cases} \tag{3}$$

where $X_f^b$ denotes the set of the finest-scale pixels in the current batch, and $\gamma = 0.999$ is the momentum decay of EMA.

Unlike multi-class classification, where FreeMatch adjusts thresholds per class to handle intra-class diversity and class adjacency, our task involves only binary classification via a sigmoid output. Thus, we simply interpret samples above the global mean $\tau_t$ as high-confidence positives, and those below $1 - \tau_t$ as high-confidence negatives. The final thresholds are:

$$\tau_h = \tau_t, \quad \tau_l = \min(1 - \tau_t, \tau_h), \tag{4}$$

with $\tau_l$ constrained to not exceed $\tau_h$.

The overall training objective combines the supervised loss on labelled samples with the semi-supervised loss on unlabelled samples, aggregated across all scales:

$$\mathcal{L} = \sum_{i \in S} \left( \mathcal{L}_i^{\text{sup}} + \mathcal{L}_i^{\text{ssl}} \right). \tag{5}$$

## 4 EXPERIMENTS

### 4.1 EXPERIMENTAL SETUP

**Training Details** We adopt RoMa (Edstedt et al., 2024) as the dense matcher and train it using the AdamW optimizer (Loshchilov, 2017), with a weight decay of 0.01. The learning rates are set to $1 \times 10^{-4}$ for the encoder and $5 \times 10^{-6}$ for the decoder. All experiments are conducted on 4 NVIDIA A5000 GPUs with a total batch size of 8. Input images are resized to $448 \times 448$, and random geometric transformations (including perspective, scaling, rotation, translation, and horizontal flipping) are applied for data augmentation on both labelled and unlabelled data. Training is run for 12,500 iterations, alternating between labelled and unlabelled batches, and requires approximately 2.5 hours per experiment.

For $\lambda^{sup}$ in Eq. (1), we follow (Edstedt et al., 2023; 2024) and set it to 0.01. For $\lambda^{ssl}$ in Eq. (2), we use $10^{-4}$, with sensitivity analysis provided in Section 5.4. The model is initialized with the MatchAnything (He et al., 2025) pretrained weights. Few-shot labelled samples are randomly drawn from the training set, while the remaining images in the training set are treated as unlabelled data. Unless otherwise specified, we use 2-shot labelled samples per dataset (see Section 5.3 for further discussion on the effect of pretrained weights and the number of labelled samples). The checkpoint achieving the best validation performance is selected as the final model, and benchmarking results are reported on the test set. All results are averaged over three runs with different few-shot splits.

**Dataset Information** We conduct experiments on four datasets spanning diverse cross-modality settings: Optical-Infrared (Li et al., 2023), Optical-Depth (Li et al., 2023), Optical-Map (Li et al., 2023), and Optical-SAR (Xiang et al., 2023). For Optical-Infrared, Optical-Depth and Optical-Map, we randomly split the data into 60%/20%/20% for training/validation/testing, while for Optical-SAR we adopt the default split. The detailed statistics of each dataset are provided in Table 1.

Table 1: Statistics of the evaluation datasets.

|  | Optical-Infrared | Optical-Depth | Optical-Map | Optical-SAR |
|---|---|---|---|---|
| Train | 120 | 120 | 120 | 2011 |
| Val | 40 | 40 | 40 | 238 |
| Test | 40 | 40 | 40 | 424 |
| Total | 200 | 200 | 200 | 2673 |

## 4.2 EXPERIMENTAL RESULTS

We benchmark against foundation models and a supervised finetuning (SFT) baseline, where the model is trained solely with the supervised loss in Eq. (1) on labelled data. The results are summarized in Table 2. We find that SFT often underperforms foundation models—except on the Optical-Map datase—primarily due to overfitting to the limited labelled samples. In contrast, the proposed SSL strategy consistently yields substantial improvements. On the Optical-Infrared dataset, our method improves $AUC$@5px by 44% over MatchAnything with only 2-shot labelled samples, underscoring the effectiveness of leveraging unlabelled data to regularize training and mitigate overfitting in the few-shot setting.

Table 2: Homography estimation on cross-modality datasets. The $AUC$ (area under curve) of the projective error in percentage is reported.

| Method | Optical-Infrared | | | Optical-Map | | | Optical-Depth | | | Optical-SAR | | |
|---|---|---|---|---|---|---|---|---|---|---|---|---|
| | @3px | @5px | @10px | @3px | @5px | @10px | @3px | @5px | @10px | @3px | @5px | @10px |
| RoMa | 1.54 | 4.48 | 8.50 | 0.00 | 0.00 | 0.00 | 1.68 | 2.85 | 5.80 | 1.76 | 5.27 | 13.18 |
| MINIMA | 8.67 | 13.16 | 19.04 | 2.88 | 7.34 | 14.79 | 3.20 | 7.48 | 17.88 | 7.38 | 22.44 | 49.45 |
| MatchAnything | 9.98 | 20.57 | 40.04 | 8.22 | 22.11 | 50.04 | 8.77 | 15.87 | 35.42 | 4.68 | 18.58 | 45.82 |
| SFT | 2.49 | 6.14 | 13.54 | 7.61 | 25.13 | 55.44 | 3.01 | 7.64 | 21.30 | 1.96 | 9.50 | 32.49 |
| Ours | **44.62** | **64.77** | **81.79** | **23.94** | **49.98** | **74.10** | **21.86** | **38.18** | **60.57** | **9.66** | **28.86** | **58.58** |

## 5 FURTHER ANALYSES

### 5.1 EFFECT OF PSEUDO-LABEL GENERATION SCHEMES

We evaluate several pseudo-label generation schemes to validate the effectiveness of the proposed method. The proposed scheme that uses the finest-scale predictions of current model as pseudo-labels is denoted as *PL-F* and compared against the following alternatives:

- *PL-T*: When the transformation between two images is known a priori (e.g., homography or camera pose), we estimate the transformation matrix from the model prediction and compute pseudo ground truth using the same procedure as in supervised learning.

- *PL*: Directly uses high-confidence predictions at each scale as pseudo-labels for loss computation at that scale.

- *PL-EMA*: Employs a teacher-student framework where the teacher model is obtained by EMA of the student model (Tarvainen & Valpola, 2017), and teacher's confident predictions serve as pseudo-labels for training the student at the corresponding scale.

- *PL-EMA-F*: A hybrid scheme that combines *PL-F* with *PL-EMA*, where the teacher's finest-scale confident predictions are downsampled to generate pseudo-labels at coarser scales for the student.

Results are shown in Fig. 2a. *PL-T* performs poorly and can even fail in cases where the geometric transformation is difficult to estimate (*e.g.*, Optical-Depth). This is likely because errors in the estimated transformation matrix amplify confirmation bias in SSL, leading to trivial solutions where the certainty branch outputs only 0s or 1s. All other schemes outperform *PL-T*. Notably, variants with the *-F* suffix consistently achieve better or comparable performance than their counterparts without the suffix (*e.g.*, *PL-F* vs. *PL*, *PL-EMA-F* vs. *PL-EMA*), validating the effectiveness of our finest-scale pseudo-labelling strategy. Between *PL-EMA-F* and *PL-F*, the latter performs better or on par, while avoiding the additional computational overhead of maintaining a teacher model. We therefore advocate *PL-F* as our final method.

### 5.2 EFFECT OF PSEUDO-LABEL THRESHOLDING SCHEMES

The parameters $\tau_h$ and $\tau_l$ in Eq. (2) control the thresholding of high-confidence predictions for pseudo-labeling. We compare different thresholding schemes. In the fixed scheme, $\tau_h$ is set to a

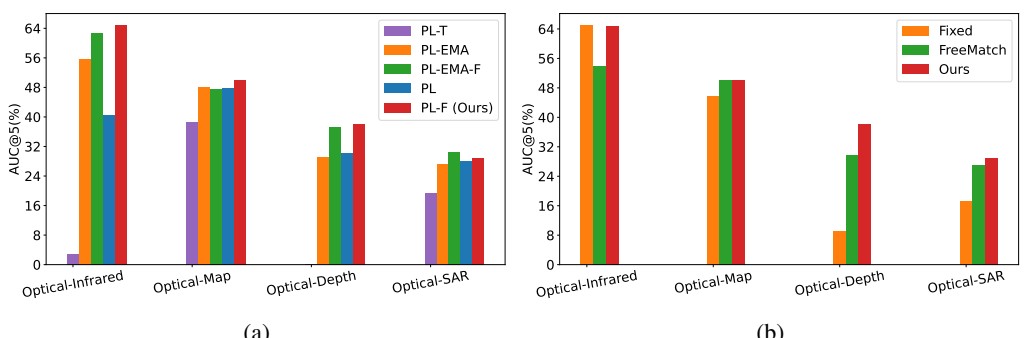

(a)                                                                    (b)

Figure 2: Effect of pseudo-label generation schemes and thresholding schemes. (a) Effect of pseudo-label generation schemes. (b) Effect of pseudo-label thresholding schemes.

constant value and $\tau_l$ is set symmetrically as $1 - \tau_h$. Following the SSL literature (Sohn et al., 2020), we set $\tau_h = 0.95$ in our experiments. FreeMatch (Wang et al., 2022) instead computes a global threshold that reflects overall model confidence and then modulates it locally based on class confidence. To adapt FreeMatch to our binary setting, we treat $(1 - \hat{p}^f(\cdot))$ as the probability for the negative class (*i.e.*, the pixel is not matchable).

Results are shown in Fig. 2b. The fixed scheme fails to perform consistently well across datasets. FreeMatch achieves performance close to ours on Optical-Map and Optical-SAR, but underperforms on Optical-Infrared and Optical-Depth. We observe that the global threshold in FreeMatch follows a trend similar to ours, gradually increasing to a high value (*e.g.*, 0.9) as training progresses. However, the additional class-wise modulation alters $\tau_h$ and $\tau_l$ in a way that appears less effective for our problem. Specifically, FreeMatch lowers the threshold for the less frequent class to encourage more pseudo-labels for that class, but this also introduces more incorrect pseudo-labels, especially when class distributions are highly imbalanced. In our case, large image overlaps result in a majority of pixels belonging to the matchable class, making this modulation detrimental and explaining why FreeMatch is less effective.

## 5.3 EFFECT OF PRETRAINED WEIGHTS

We evaluate three publicly available pretrained models: RoMa (Edstedt et al., 2024), which is pre-trained only on single-modality image pairs; MINIMA (Ren et al., 2025), which adopts the RoMa architecture and employs a two-stage pretraining pipeline (*i.e.*, first on single-modality data and then on large-scale synthetic cross-modality data); and MatchAnything (He et al., 2025), which also builds on the RoMa architecture but is pretrained directly on large-scale synthetic cross-modality data. Intuitively, MINIMA and MatchAnything should provide stronger initialization for downstream cross-modality image matching tasks, as their pretraining already incorporates cross-modality supervision.

Experimental results with different pretrained weights are shown in Fig. 3. The proposed SSL framework consistently outperforms the SFT counterpart across different pretrained weights and varying numbers of labelled samples. SSL with MatchAnything pretrained weights generally yields the best performance when only a small number of labelled samples are available, although its advantage diminishes when the number of labelled samples increases (*e.g.*, in the 10-shot setting). Overall, these results highlight that cross-modality foundation models provide the greatest benefit in the extreme low-label regime.

## 5.4 SENSITIVITY ON $\lambda^{ssl}$

The parameter $\lambda^{ssl}$ in Eq. (2) controls the relative contribution of the certainty prediction branch. Experimental results with varying $\lambda^{ssl}$ are shown in Fig. 4. We find that model performance remains stable within the range $[0, 10^{-4}]$ across all evaluation datasets, whereas excessively large values (*e.g.*, $10^{-2}$) degrade performance, particularly under low-shot setting. This behavior may be attributed to the fact that RoMa's regression and certainty branches share the same input features and differ only in the prediction head; thus, backpropagation from the regression loss alone can also learn useful representations. Furthermore, since pseudo-labels are inherently noisy, assigning a large

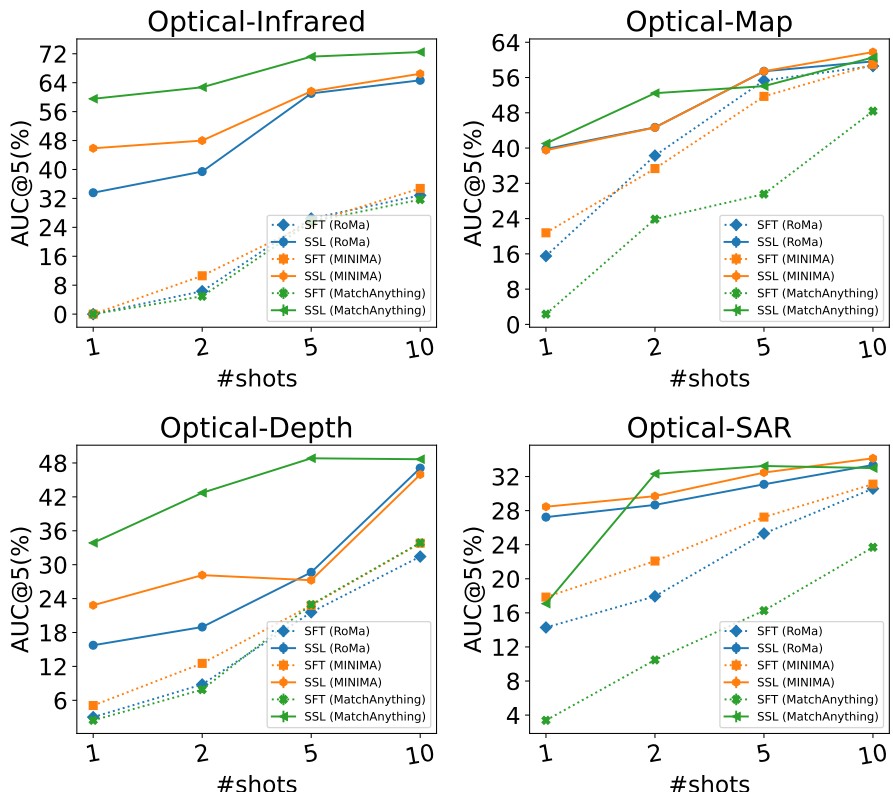

Figure 3: Effect of pretrained weights. SFT denotes the supervised finetuning baseline, while SSL refers to the proposed semi-supervised learning method. The names in brackets indicate the pretrained weights used for model initialization. Across different pretrained weights and varying numbers of labelled samples, SSL consistently outperforms its SFT counterpart.

weight to the certainty loss may amplify confirmation bias—where errors in pseudo-labels reinforce themselves—often leading to trivial solutions.

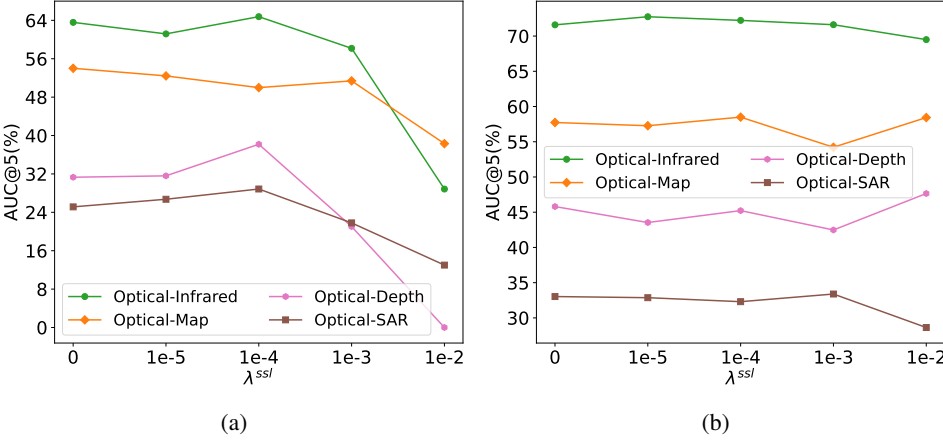

Figure 4: Sensitivity on $\lambda^{ssl}$. (a) Trained with 2-shot labelled samples. (b) Trained with 10-shot labelled samples. We find that model performance remains stable within the range $[0, 10^{-4}]$ across all evaluation datasets, whereas excessively large values (*e.g.*, $10^{-2}$) lead to performance degradation, particularly under low-shot setting.

## 5.5 EFFECT OF DATASET DIFFICULTY

Pseudo-labeling builds on the self-training assumption in SSL, namely that high-confidence predictions are more likely to be correct. However, as variations between two images grow larger and more complex, obtaining reliable pseudo-labels becomes increasingly difficult. To study the effect of dataset difficulty on our method, we augment the Optical-Map dataset with geometric transformations and Gaussian noise (Hendrycks & Dietterich, 2019) of increasing magnitude. Results in Fig. 5 show that foundation models are highly sensitive to Gaussian noise, with matching accuracy quickly dropping to zero, while SFT with 2-shot supervision also performs poorly. By contrast, our method consistently outperforms foundation models and SFT, demonstrating that finetuning with the proposed pseudo-labeling strategy is effective even under challenging conditions.

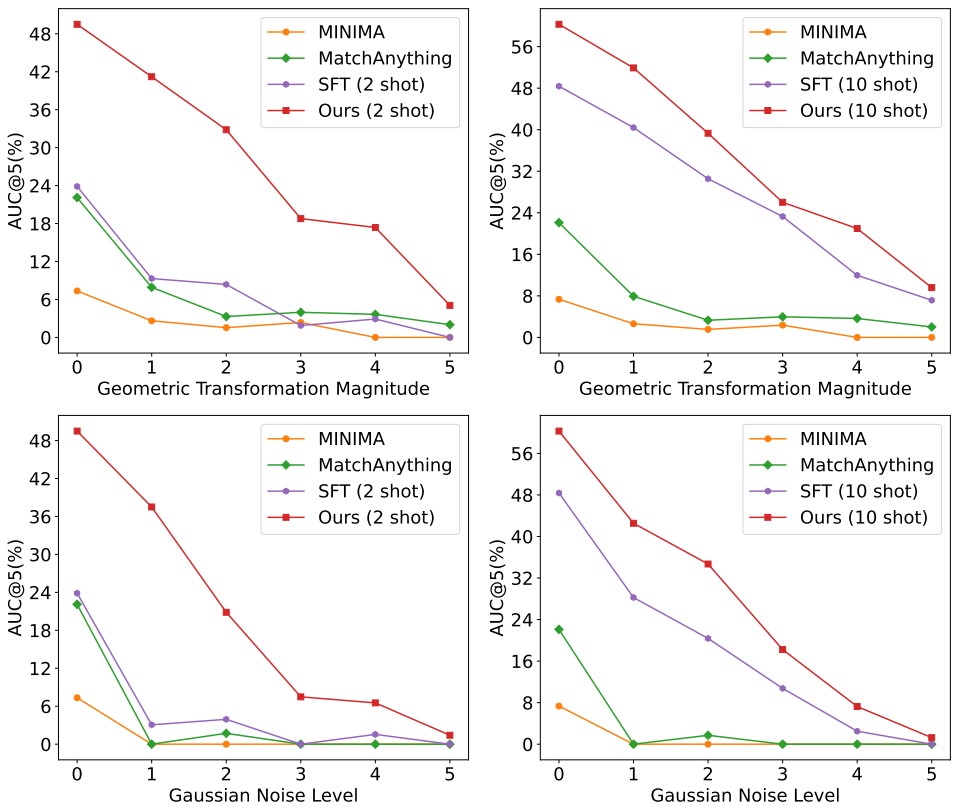

Figure 5: Effect of dataset difficulty on the Optical-Map dataset. The dataset is perturbed by geometric transformations or Gaussian noise of increasing magnitude (with 0 denoting the unaltered dataset). Our method consistently outperforms foundation models and SFT under challenging conditions.

## 6 CONCLUSIONS

In this work, we propose a semi-supervised framework for adapting pretrained dense matchers to downstream cross-modality data by leveraging a small number of labelled samples together with abundant unlabelled samples. Specifically, we exploit the inherent multi-scale structure of dense matchers and use the finest-scale predictions as pseudo-labels to supervise learning at coarser scales. We benchmark the proposed method against foundation models and demonstrate consistent and substantial improvements across diverse cross-modality datasets. Remarkably, with only 2-shot labelled samples, our approach improves $AUC@5$px by up to 40% compared to foundation models. We further validate the robustness of our framework under different initialization weights and varying numbers of labelled samples. Finally, we examine the effect of dataset difficulty by introducing geometric transformations and Gaussian noise of increasing severity, and observe that our method consistently outperforms both supervised finetuning and foundation models under these challenging conditions.

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
