# OpenReview forum: "Towards Few-Shot Adaptation for Dense Cross-Modality Image Matching"
_ICLR.cc/2026/Conference — ICLR 2026 Conference Withdrawn Submission_

### Official Review · Reviewer_oTj9 · 2025-10-27

**Soundness:** 2
**Presentation:** 3
**Contribution:** 2
**Rating:** 2
**Confidence:** 3

**Summary:**

This work proposes a few-shot framework for enabling cross-modality matching. The key idea is to have few-shot labeled data plus unlabeled data for semi-supervised learning. It uses fine-grained high-confidence pseudo-labels to supervise course-grained predictions.

**Strengths:**

The method is simple.
Experiments shows that it can improve the performance with tiny scale datasets.

**Weaknesses:**

The method is too simple, which lacks algorithm novelty. The key idea is the standard semi-supervised learning. Where it assumes a small portion of the data are labeled and another bigger portion of the data are unlabled, and conduct confidence-based semi-supervised learning. Tricks like using fine-grained high confidence region as pseudo-labels are very similar to semi-supervised learning domain where some data augmentations are applied to make the predictions of the same model capable of providing meaningful signals.

The scale of the experiment is tiny. All datasets used in the paper only contain <2k images, where labeling on such scale is arguably much easier than designing complex methods for pseudo-labeling.

It compares to other cross modality methods like MatchAnything, however, it does not compare over the data that the baseline used. And it finetunes from the MatchAnything checkpoint. These make the comparison unfair.

**Questions:**

NA

---

### Official Review · Reviewer_KLQ1 · 2025-10-29

**Soundness:** 2
**Presentation:** 2
**Contribution:** 2
**Rating:** 4
**Confidence:** 4

**Summary:**

This paper addresses the problem of few-shot adaptation for dense cross-modal matching—the task of aligning images from different modalities where significant differences exist in terms of appearance and geometry. The key challenge is to leverage limited labeled data while utilizing large amounts of unlabeled data, effectively transferring a pre-trained model’s knowledge to new domains. The authors propose a novel approach that internally uses the finest scale of the model as a "teacher" to guide the coarse scales, without requiring a separate teacher model, and they introduce self-supervised learning for fine-tuning. Experiments demonstrate the effectiveness of proposed method.

**Strengths:**

1. The paper introduces a multi-scale self-supervised training strategy that adapts dense cross-modal matching models with very limited labeled data.

2. The pseudo-labeling mechanism is well aligned with the cross-modal setting and reduces reliance on annotations.

3. The experimental design is solid, covering multiple cross-modal datasets (e.g., optical–infrared, depth) and demonstrating cross-domain generality.

**Weaknesses:**

1. Limited novelty relative to prior semi-/self-supervised dense matching: While the combination of multi-scale training and pseudo-labels is reasonable, the methodological contribution feels incremental for a venue like ICLR.

2. Writing quality and typos: There are several minor errors (e.g., “scales..”).

3. Inconsistent hyphenation and terminology: Mixed usage such as “finest-scale” vs. “fine-level”, “pseudo-labels” vs. “pseudo labels”, and “coarser-scale” vs. “coarser scales.”

4. Sensitivity of the self-supervised loss weight (λ_ssl): The paper points out that λ_ssl is "basically stable" in the interval [0, 10^-4], but degrades significantly when the value is 10^-2, especially with low sample sizes. This means that in practical deployments, the hyperparameter is very sensitive to "excessive" values, and the tuning burden and robustness need further verification.

5. Backbone generality: Results are shown primarily with RoMa. It remains unclear whether the approach transfers to other dense matchers with different multi-scale designs (e.g., DKM variants or more recent architectures).

**Questions:**

See the weaknesses.

---

### Official Review · Reviewer_XESp · 2025-10-31

**Soundness:** 3
**Presentation:** 3
**Contribution:** 3
**Rating:** 4
**Confidence:** 4

**Summary:**

The paper proposes a new approach to few-shot dense matching across image modalities. The idea is to adapt a matcher (like RoMa, which was pretrained in a unimodal setting, or more recent methods adapted to synthetic multi-modal data) to a particular cross-modal setting (e.g., matching optical and infrared image pairs). The method is termed few-shot since it assumes a small number of labeled (solved) pairs in the target domain, along with an abundant number of unlabeled pairs. The unlabeled pairs are used in a self-supervised manner, based on the predictions of the matcher at the finest resolution in the matching hierarchy, with a mechanism to account for confidence at intermediate levels. The method is evaluated in comparison to vanilla fine-tuning on the support samples, as well as to other foundation-model matchers that work in the zero-shot setting.

**Strengths:**

1) Like the authors, I am not aware of previous semi-supervised adaptation of dense matchers across domains, and I believe this is an interesting and natural setting that is worth studying.
2) The proposed method is quite simple and shows strong improvements, mainly in that it is able to successfully exploit the available unlabeled examples.
3) The idea of using pseudo-labels in a confidence-aware hierarchical bottom-up manner is particularly compelling.
4) Generally well written, especially the background and motivation.

**Weaknesses:**

1) The setting is somewhat specific in that it assumes the availability of *both* (a few) labeled and (many) unlabeled image pairs. Few-shot settings do not typically assume abundant unlabeled data. More importantly, since the method primarily fine-tunes using the unlabeled examples, it is unclear why (and whether) the few labeled examples are necessary in this setup.
2) A main component of the method, the dynamic threshold mechanism for determining pseudo-label confidence, is only briefly presented in Eq. (3) and is not explained, nor are its training dynamics illustrated.
3) Experiments are limited to settings where a 2D mapping (homography) exists between the views. It would be more informative and convincing to include datasets with general 3D, non-planar scenes.
4) In the comparisons, it should be made very clear that the proposed method uses more information: The prior methods operate in a zero-shot manner, and the SFT baselines use only the labeled support set.
5) Visualization is limited in the number and type of examples. Prior works often include more visualizations and show matching lines with color coding that convey accuracy or confidence.

**Questions:**

1) It would be useful to include the ablation where no labeled data is used at all (i.e., add the 0-shot point to Figure 3).
2) In Figure 3, the SFT (dashed) results are confusing: when the shot number drops to 1, the AUC drops near zero (e.g., for MatchAnything), even though the zero-shot performance in Table 2 is much higher. I would expect performance to improve rather than degrade when adding labeled examples.
3) Did you run experiments on more general non-homography 3D settings? More broadly, why were these datasets chosen instead of others used in prior works (e.g., MINIMA, MatchAnything)?
4) Could you provide more detailed visualizations, for example with color-coded matching lines? The current visualizations focus on confidence but do not convey matching accuracy.

---

### Official Review · Reviewer_Ns2n · 2025-11-03

**Soundness:** 3
**Presentation:** 3
**Contribution:** 2
**Rating:** 6
**Confidence:** 2

**Summary:**

The paper addresses the problem of adapting pretrained dense cross-modality matchers to new modalities under few-shot conditions. The idea of leveraging fine-scale predictions as pseudo-labels for coarser scales within a semi-supervised framework is empirically effective.

**Strengths:**

The proposed framework is conceptually simple yet effective, making good use of the multi-scale structure of dense matchers without introducing heavy additional modules. And the empirical results are strong and consistent across multiple settings.

**Weaknesses:**

1. The authors state that “using the finest-scale predictions to supervise coarser-scale learning”. However, the conclusion that fine-scale predictions are inherently more reliable and thus suitable for supervising coarser levels appears somewhat direct and is not empirically verified. It would be important to show evidence that fine-scale predictions are indeed more accurate, for instance by analyzing their confidence-accuracy correlation.
(2) Similarly, the proposed “fine-to-coarse” mechanism reverses the common coarse-to-fine optimization paradigm. The paper does not clearly justify why this reverse direction should be beneficial or how it interacts with the hierarchical feature representations. A clearer theoretical or empirical rationale would strengthen the method’s foundation.
(3) While the method shows substantial overall gains compared to baselines, Figure 3 indicates that performance changes little when increasing the number of labeled samples from 2-shot to 10-shot. This trend deserves explanation, as it may suggest that the model quickly saturates with very few labels or that the semi-supervised loss dominates the learning process.
(4) The definition of the few-shot setting is somewhat ambiguous. The paper mentions “2-shot,” but it is unclear how a “shot” is defined. It would help to clarify whether an all-way or fixed-pair sampling strategy was used.
(5) Although several pseudo-label generation schemes are compared, the paper does not provide quantitative or visual evaluation of pseudo-label quality.
(6) The paper would also benefit from a discussion of possible limitations or failure cases.
(7) From the perspective of innovation, the work extends existing semi-supervised learning concepts to dense cross-modality matching and introduces a fine-to-coarse pseudo-labeling strategy. While the approach achieves notable empirical improvements, its novelty lies primarily in adapting known SSL principles to a new context rather than introducing a new paradigm. The contribution may be incremental but practically valuable for few-shot cross-modality adaptation.

**Questions:**

pls refer to the weakness

---

### Note · Authors · 2025-11-13

I have read and agree with the venue's withdrawal policy on behalf of myself and my co-authors.